# Structural basis of cell wall anchoring by SLH domains in *Paenibacillus alvei*

Ryan J. Blackler [1,4], Arturo López-Guzmán[2], Fiona F. Hager[2], Bettina Janesch[2], Gudrun Martinz[3], Susannah M.L. Gagnon[1], Omid Haji-Ghassemi[1,5], Paul Kosma [3], Paul Messner[2], Christina Schäffer[2] & Stephen V. Evans[1]

Self-assembling protein surface (S-) layers are common cell envelope structures of prokaryotes and have critical roles from structural maintenance to virulence. S-layers of Gram-positive bacteria are often attached through the interaction of S-layer homology (SLH) domain trimers with peptidoglycan-linked secondary cell wall polymers (SCWPs). Here we present an in-depth characterization of this interaction, with co-crystal structures of the three consecutive SLH domains from the *Paenibacillus alvei* S-layer protein SpaA with defined SCWP ligands. The most highly conserved SLH domain residue SLH-Gly29 is shown to enable a peptide backbone flip essential for SCWP binding in both biophysical and cellular experiments. Furthermore, we find that a significant domain movement mediates binding by two different sites in the SLH domain trimer, which may allow anchoring readjustment to relieve S-layer strain caused by cell growth and division.

[1] Department of Biochemistry and Microbiology, University of Victoria, Victoria, BC V8W 3P6, Canada. [2] Department of NanoBiotechnology, NanoGlycobiology Unit, Universität für Bodenkultur Wien, 1190 Vienna, Austria. [3] Department of Chemistry, Universität für Bodenkultur Wien, 1190 Vienna, Austria. [4] Present address: Zymeworks Inc., Vancouver, BC V6H 3V9, Canada. [5] Present address: Department of Biochemistry and Molecular Biology, University of British Columbia, Vancouver, BC V6T 1Z3, Canada. These authors contributed equally: Ryan J. Blackler, Arturo López-Guzmán. Correspondence and requests for materials should be addressed to C.Säf. (email: christina.schaeffer@boku.ac.at) or to S.V.E. (email: svevans@uvic.ca)

Prokaryotic cell envelopes often include proteinaceous surface (S-) layers composed of self-assembling subunits in porous two-dimensional crystalline lattices[1–7]. S-layers can have a variety of critical functions including structural maintenance, environmental protection, adhesion, filtering, and virulence[8–14]. These characteristics make S-layers attractive targets for new antibiotics and provide biotechnological potential for the display of functional epitopes with precise geometries in vaccine design, drug delivery, molecular electronics, and filtration[15–23]. Understanding their three-dimensional structure is key to exploiting the therapeutic or biotechnological potential of S-layers.

The maintenance of S-layers in biological contexts depends on stable and flexible attachment to constantly changing cell surfaces. S-layers have evolved covalent and non-covalent anchor mechanisms to a variety of cell wall components, including the archaeal plasma membrane[22,24,25], the peptidoglycan (PG) layer of Gram-positive bacteria[9,22,26,27], and the lipopolysaccharide of Gram-negative bacteria[28–30].

S-layer anchoring is best characterized for Gram-positive bacteria, where S-layer proteins non-covalently attach to the PG layer via species- and strain-specific non-classical secondary cell wall polymers (SCWPs)[26]. Non-classical SCWPs are neutral or anionic polysaccharides not classifiable as teichoic or teichuronic acids (classical SCWPs), and are covalently linked to muramic acid residues of PG, at least in some cases through phosphodiester bonds. SCWP structures have been reviewed in detail elsewhere[8].

The most common, but not exclusive, means of SCWP binding is through S-layer homology (SLH) domains, which are often present in triplicate at the termini of S-layer and other extracellular proteins[4,22,30–33]. Within the conserved protein domain family SLH (pfam00395)[34], there are 14,079 sequences identified as SLH domains across 651 bacterial species, widely in the phyla Firmicutes (378 species), Cyanobacteria (100 species), Proteobacteria (41 species), and Actinobacteria (33 species), among others, suggesting the early evolution of the SLH–SCWP interaction for anchoring proteins to the surfaces of bacteria[35].

The interaction of SLH domains with several SCWPs depends on a common ketal–pyruvate modification of SCWPs that imparts a negative charge[35–37]. SLH domains contain a highly conserved TRAE motif (SLH domain residues 42–45, Fig. 1) where the arginine residue has been shown to be critical to PG binding[27,31,38], suggesting that it interacts with the negatively charged pyruvate moiety of these SCWPs.

The crystal structure of the SLH domain trimer from the B. anthracis S-layer protein Sap revealed the three domains arranged in approximate threefold symmetry, where each SLH domain contributes one helix to a parallel three-helix bundle core and a second helix rotated ~90° from the core to form three lobes[39]. The conserved TRAE motifs are located at the N-termini of the core helices adjacent to three grooves, suggesting that these grooves are the sites for SCWP binding. The structure of B. anthracis SCWP was elucidated subsequently[40], and a crystal structure of the SLH domains of Sap in complex with a synthetic trisaccharide approximating the terminal repeat of B. anthracis SCWP was also recently reported[41]. This structure revealed ligand bound in only one of the three grooves and confirmed that SLH-Arg43 of the conserved TRAE motif interacts with the pyruvate moiety of SCWP. However, questions remain as to the functionality of the remaining two grooves and the roles of additional conserved residues.

To investigate the molecular details of SCWP binding by SLH domains in depth, we selected the Paenibacillus alvei CCM 2015[T] S-layer protein SpaA for structural studies. SpaA possesses three consecutive SLH domains near its N-terminus, followed by a large C-terminal region presumed to harbor the self-assembling domain(s)[5,27,42]. The SCWP from P. alvei CCM 2051[T] has been fully defined (Fig. 2)[8,37,43,44], which enables an investigation of the precise interactions between SLH domains and SCWP.

SpaA possesses the conserved TRAE motif in its first SLH domain (SLH1) and variants of this motif, TVEE in its second (SLH2), and TRAQ in its third (SLH3) (Fig. 1). These motifs were shown to contribute unequally to SCWP binding, where their mutation to TAAA resulted in 37%, 88%, and 50% of wild-type (wt) cell wall binding in pelleting assays, respectively[27,45]. These variations of the conserved TRAE motif make SpaA an excellent model to investigate its role in SCWP binding. Similar inequality within SLH domain repeats has been observed in S-layer proteins from other organisms[46,47], but there is no obvious relationship between primary sequence and SCWP-binding stoichiometry or affinity and a structural foundation is necessary to shed light on evolved differences in SCWP-binding mechanisms.

Here we report crystal structures of the SLH domains of SpaA, unliganded and in complex with synthesized monosaccharide and disaccharide building blocks of P. alvei CCM 2051[T] SCWP. These structures reveal the contributions of many conserved residues to SCWP binding, notably including the most highly conserved residue SLH-Gly29, which is confirmed by additional co-crystal structures, thermodynamic binding analysis, and microbiological functional assays for SLH-Gly29Ala mutants. Together, these data provide novel molecular insights into this biologically important interaction.

## Results

**SpaA_SLH structure and SCWP binding.** The crystal structure of SpaA_SLH reveals an overall fold similar to the three-pronged spindle

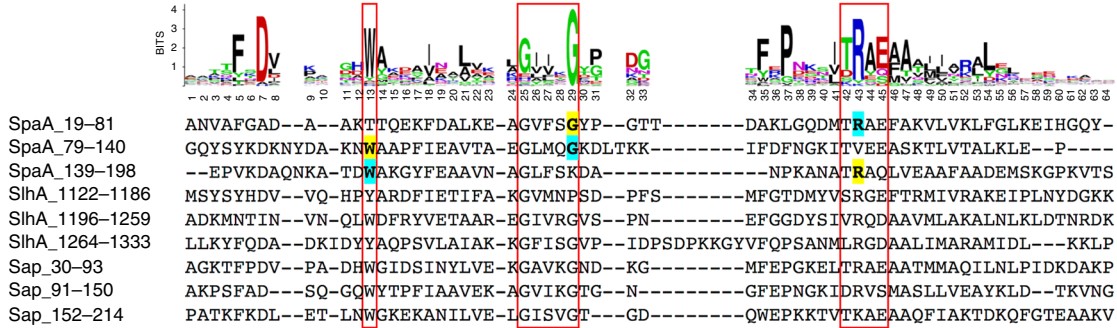

**Fig. 1** SLH domain sequence alignments. Sequence logo of the SLH domain profile from PROSITE #PS51272, with the sequences of the SLH domains from P. alvei SpaA, SlhA, and B. anthracis Sap aligned underneath. Throughout the manuscript, references to conserved residues of the SLH domain profile are preceded by "SLH-" to differentiate from SpaA_SLH amino acid labels. The conserved SLH-Trp13, GIIxG motif, and TRAE motif are boxed in red. The conserved SLH-Trp13, SLH-Gly29, and SLH-Arg43 that contribute to ligand binding in SpaA grooves 1 and 2 are highlighted in yellow and cyan, respectively

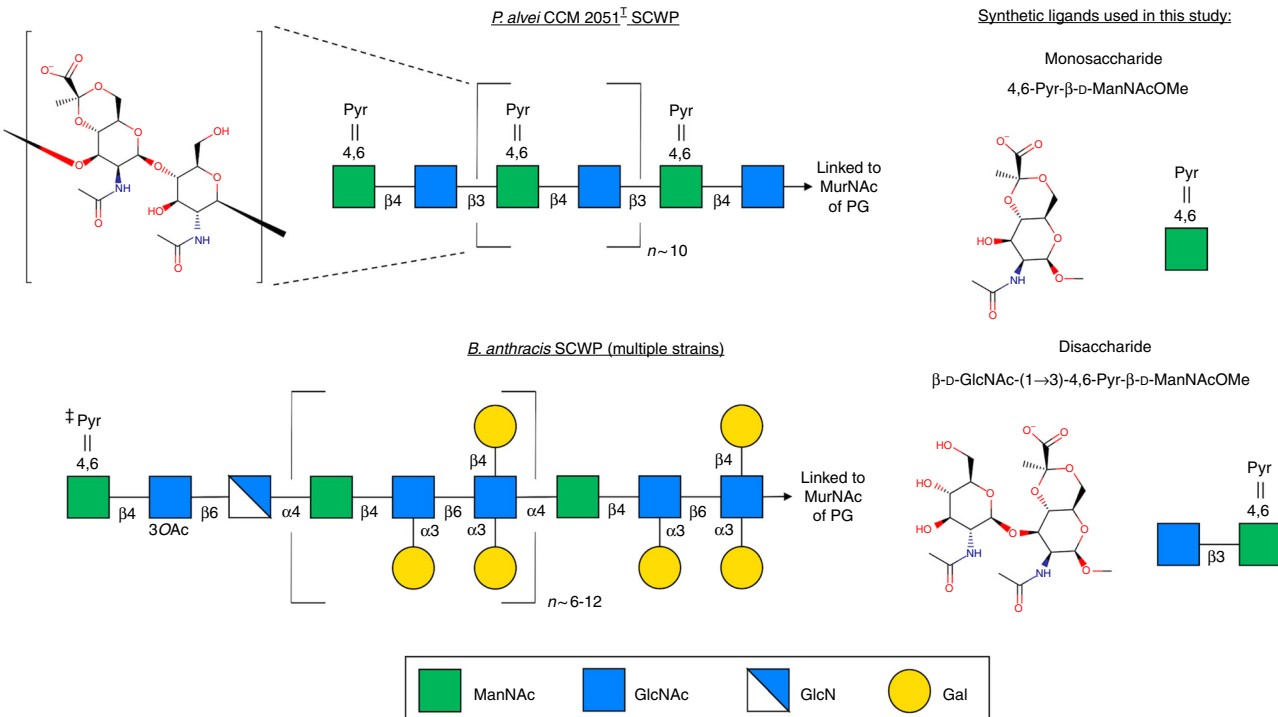

**Fig. 2** SCWP structures and synthetic ligands used in this study. Schematic diagrams of the SCWP from *P. alvei* CCM 2051[T] and *B. anthracis* (multiple strains) are shown on the left, with a detailed chemical diagram of the repeating unit shown for the former[8, 37, 40, 43, 44, 61]. The synthetic ligands used in this study, representing building blocks of *P. alvei* CCM 2051[T] SCWP, are shown on the right. ‡Pyruvylation, *O*-acetylation, and *N*-deacetylation of the terminal trisaccharide repeat were determined for *B. anthracis* Ba684[40]. There remains some uncertainty in the extent of pyruvylation of some *B. anthracis* SCWPs, because the use of HF to cleave SCWPs from PG also liberates the acid-sensitive ketal groups[8]

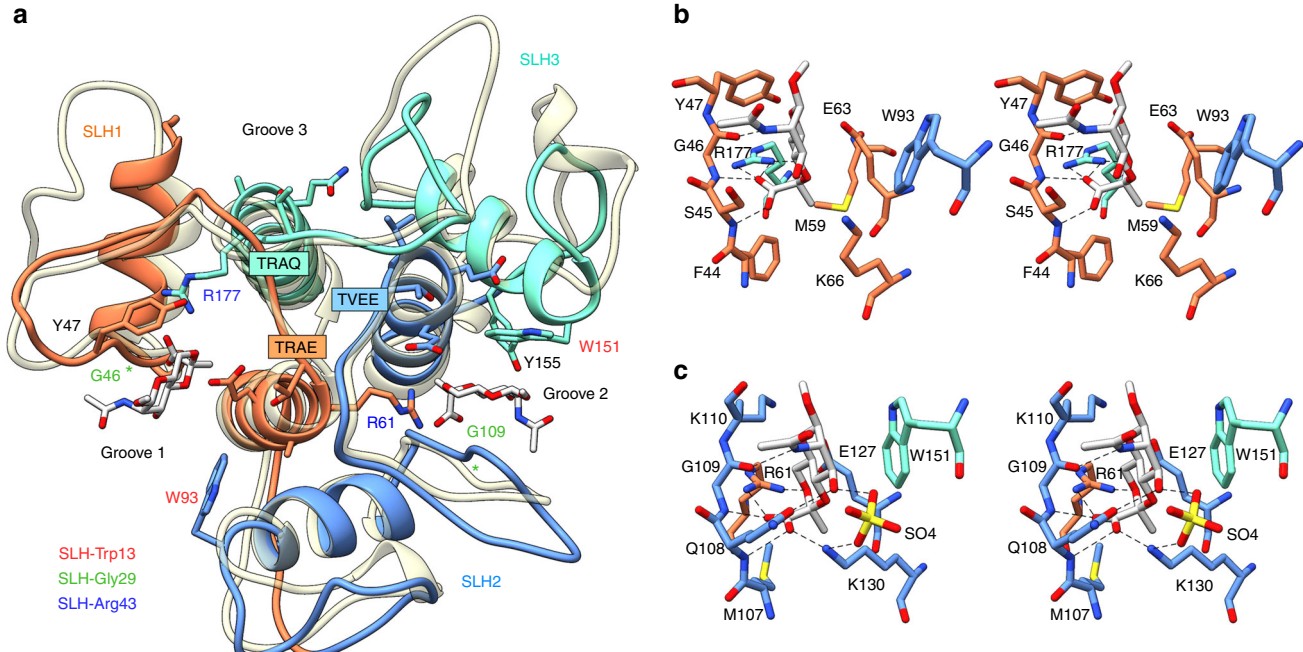

**Fig. 3** SpaA$_{SLH}$ structure and SCWP ligand binding. **a** Ribbon diagram of SpaA$_{SLH}$ with bound monosaccharide (PDB 6CWI; Supplementary Table 1) and *B. anthracis* Sap (PDB 3PYW) overlaid in tan. SpaA$_{SLH}$ is colored with SLH1 orange, SLH2 blue, and SLH3 aquamarine. Residue labels are colored by conserved SLH domain residue (legend bottom left). Stereo views of monosaccharide binding in G1 and G2 are shown in **b** and **c**

described for the SLH domains of *B. anthracis* Sap[39] (Fig. 3a, Supplementary Fig. 1, and Supplementary Table 1). The TRAE, TVEE, and TRAQ motifs of SpaA are located at the N-termini of the core helices adjacent to three grooves (labeled here G1, G2, and

G3). The TRAE and TRAQ motifs each contribute to two of the three grooves, as their arginine residues thread under the connecting loops of the adjacent SLH domains and into the neighboring grooves. In the case of SLH2, Val125 of the TVEE motif

**Table 1 ITC analyses of 4,6-Pyr-β-D-ManNAcOMe binding**

|  | $-T\Delta S$ (kJ/mol) | $\Delta H$ (kJ/mol) | $\Delta G$ (kJ/mol) | Stoichiometry | $K_A$ (M$^{-1}$) | $K_D$ (nM) |
|---|---|---|---|---|---|---|
| SpaA$_{SLH}$ | 45.74 ± 16.01 | −87.85 ± 15.62 | −42.10 ± 0.65 | 0.91 ± 0.04 | $3.48 \times 10^7$ ± 0.36 | 29 |
| SpaA$_{SLH}$/G109A | 13.33 ± 7.0 | −50.64 ± 7.29 | −37.31 ± 0.33 | 0.92 ± 0.04 | $4.48 \times 10^6$ ± 0.62 | 226 |
| SpaA$_{SLH}$/G46A/ G109A | No binding |  |  |  |  |  |

corresponds in position to the conserved SLH-Arg43 of the TRAE and TRAQ motifs but does not protrude into the neighboring G3. The final residue of each motif, Glu63, Glu127, or Gln179, lines the groove beside its parent helix.

To investigate the molecular details of SCWP binding by SLH domains, we synthesized the monosaccharide 4,6-Pyr-β-D-ManNAcOMe (see Supplementary methods and Supplementary Figs. 2–4), which represents the pyruvylated moiety of *P. alvei* SCWP (Fig. 2), and co-crystallized it with SpaA$_{SLH}$. The first co-crystal structure determined reveals ligand bound in G2 only, in a narrow pocket formed by conserved residues from all three SLH domains: Arg61 (corresponding to the conserved SLH-Arg43 of the TRAE motif, Fig. 1) from SLH1; Met107, Gln108, Gly109 (corresponding to the conserved SLH-Gly29), Lys110, Glu127, and Lys130 from SLH2; and Trp151 (corresponding to the conserved SLH-Trp13) from SLH3 (Figs. 1 and 3). The pyruvate moiety of the ligand is bound deep in the pocket through salt bridge interactions to Arg61 and Lys130 and hydrogen bonds to Gln108 and Gly109 backbone amides, while the hydrophobic face of the ManNAc ring stacks against Trp151.

In the initial liganded and unliganded structures, multiple conformations were observed for residues 44–54 that form the connecting loop between the two helices of SLH1 beside G1 (Supplementary Fig. 5). The loop makes unique crystal contacts in each case and has mean B-factors elevated between 4 and 49% of the molecule mean. We investigated the effect of this flexibility on potential ligand binding in G1 by solving additional co-crystal structures in different space groups (*P*1 and *C*2, Supplementary Table 1), where further unique conformations of residues 44–54 were observed. Ligand binding in G2 is the same in all cases except for the *C*2 structure (PDB 6CWI), which shows a sulfate ion coordinated to Lys130 and ligand in G2. Only the *C*2 structure displays electron density corresponding to a partially occupied site in the G1 pocket of molecule A (out of two molecules A and B in the asymmetric unit [AU]), with G1 of molecule B occluded by molecule A (Supplementary Fig. 6). Nevertheless, this weakly bound 4,6-Pyr-β-D-ManNAcOMe in G1 displays similar interactions as the ligand bound in G2 (Fig. 3). The pyruvate moiety is coordinated by a salt bridge from Arg177 and forms hydrogen bonds to Ser45 N and Gly46 N, which are equivalent to residues Arg61, Gln108, and Gly109 in G2. In this case, no stacking interactions are made from Trp93 (equivalent to Trp151 in G2).

To further investigate the activity of these two potential binding sites, we performed isothermal titration calorimetry (ITC) analysis of SpaA$_{SLH}$ binding to 4,6-Pyr-β-D-ManNAcOMe in solution, which revealed 1:1 binding with an apparent dissociation constant ($K_D$) of 29 nM (Table 1 and Supplementary Fig. 7). This 1:1 binding in solution likely occurs in G2, because this site is occupied in all five crystallographically unique molecules and has better complementarity with ligand than does G1, which was only occupied in one of five molecules. Therefore, the monosaccharide binding observed in G1 is likely to be an artifact of crystallization that does not occur in solution.

**SpaA binds a terminal non-reducing end epitope of SCWP.** To further refine the identity of the SCWP epitope bound by

SpaA$_{SLH}$, we synthesized the disaccharide β-D-GlcNAc-(1 → 3)-4,6-β-D-ManNAcOMe for co-crystallization and binding analysis (Fig. 2). This disaccharide represents an internal repeat of *P. alvei* SCWP as opposed to the terminal non-reducing end disaccharide Pyr-β-D-ManNAcOMe-(1 → 4)-β-D-GlcNAc, with the former being more tractable to synthesis. The internal disaccharide showed no binding to SpaA$_{SLH}$ by ITC (Supplementary Fig. 7), which suggests that SpaA$_{SLH}$ is specific for an epitope including the terminal non-reducing end 4,6-Pyr-β-D-ManNAc residue of SCWP rather than an internal epitope.

Despite the lack of binding measured by ITC, a co-crystal structure with the internal disaccharide was obtained, again indicating that binding is enhanced significantly in the crystalline environment compared to in solution. In both molecules in the AU of this structure, the 4,6-Pyr-β-D-ManNAcOMe moiety of the disaccharide is bound in the G2 pocket in a manner similar to the monosaccharide ligand (Supplementary Fig. 8). The two molecules display fragmented electron density for the GlcNAc moiety in two different conformations, neither of which form hydrogen bonds to the protein (Supplementary Fig. 6). Furthermore, SpaA$_{SLH}$ residues 139–152 are disordered in one of two molecules in the AU, whereas they are ordered in all other structures of SpaA$_{SLH}$. This region includes Trp151 (the conserved SLH-Trp13), which normally stacks against the 4,6-Pyr-β-D-ManNAcOMe moiety. In the molecule where this loop is disordered, the GlcNAc moiety of the disaccharide occupies the expected location of Trp151, thus suggesting that this region becomes disordered to avoid a clash with the disaccharide ligand.

**SLH-Gly29 enables a backbone flip required for SCWP binding.** In addition to the TRAE motif, SLH domains contain a highly conserved GIIxG motif (residues 25–29, Fig. 1), where the second glycine (SLH-Gly29) is the most conserved residue of the SLH domain profile[31,48]. SpaA possesses variants of this motif, with GVFSG, GLMQG, and GLFSK present in SLH1–SLH3. These motifs are located in loops between the two helices of each SLH domain that line one side of each groove (Figs. 1 and 3). The first G of each motif is within a turn exiting α1 and transitioning into this loop, and the hydrophobic residues VF, LM, and LF are oriented toward the hydrophobic core of each SLH domain lobe. In both G1 and G2, the non-conserved SLH-28 residue is observed to make a hydrogen bond from its backbone carbonyl to the side chain of the conserved SLH-Arg43 of the TRAE motif.

A remarkable series of structural transformations involving SLH-28 and SLH-Gly29 is observed with SCWP binding. Upon binding in G2 of *wt* SpaA$_{SLH}$, the phi angle of Gly109 (SLH-Gly29) and the psi angle of Gln108 (SLH-28) each flip by ~180° (Fig. 4 and Supplementary Table 2). The backbone carbonyl of Gln108, which hydrogen bonds to Arg61 (the conserved SLH-Arg43) when unliganded, is resultantly replaced in space by the Gly109 backbone amide nitrogen, which forms a hydrogen bond to a pyruvyl carboxyl of ligand. Concurrently, the removal of the hydrogen bond between Gln108 and Arg61 allows Arg61 to make a bidentate interaction with a pyruvyl carboxyl and O6 of ligand.

The hydrogen bond between the SLH-28 backbone carbonyl and SLH-Arg43 in the unliganded state may support the proper folding of the SLH domain trimer, as these residues precisely interlock

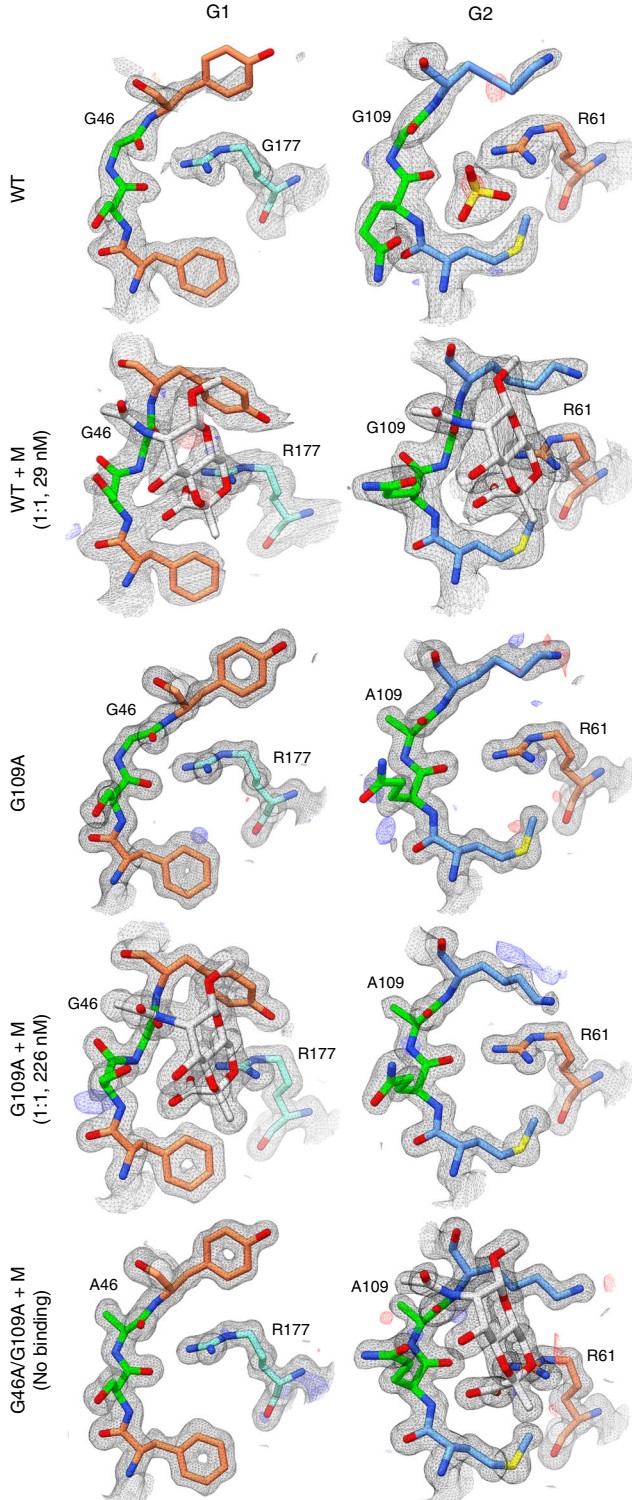

**Fig. 4** SLH-Gly29 backbone flip. 2Fo–Fc electron density maps contoured to 1σ (black) and Fo–Fc maps contoured to 3σ (blue) and −3σ (red) are shown for residues SLH-27 to SLH-30 and SLH-Arg43 in groove 1 (G1) and groove (G2) of SpaA$_{SLH}$ structures. Residues are colored according to SLH domain as in Fig. 3, with SLH-28 and SLH-29 in green to highlight the backbone flip. WT wild-type SpaA$_{SLH}$, M monosaccharide ligand, G109A SpaA$_{SLH}$/G109A single mutant, G46A/G109A SpaA$_{SLH}$/G46A/G109A double mutant. Monosaccharide ligand and corresponding electron density are shown for structures where density was observed, and binding stoichiometry and $K_D$ as determined by ITC are given

between neighboring SLH domains (Fig. 3). This is consistent with our finding that the TRAE motif is important for the overall folding of the SLH domain trimer, as attempts to express SpaA$_{SLH}$/TAAA/ TAAA double mutants resulted in insoluble inclusion bodies (see Supplementary Note 1). However, the presence of this stabilizing hydrogen bond between the SLH-28 backbone carbonyl and SLH-Arg43 would prevent SLH-Arg43 from making an ideal bidentate interaction with ligand, and ligand would be further deprived of the hydrogen bond from the backbone amide of SLH-Gly29. The backbone flip requires the flexibility of SLH-Gly29 in this location as the resulting phi–psi angles are disallowed for all other amino acids. Therefore, the near universal conservation of SLH-Gly29 suggests that this backbone flip is part of a conserved mechanism for SLH–SCWP binding.

To test our hypothesis that any other amino acid would prevent the backbone flip and impede binding, a SpaA$_{SLH}$/G109A mutant was constructed for analyses. ITC revealed 1:1 binding to monosaccharide 4,6-Pyr-β-D-ManNAcOMe with 226 nM affinity, nearly an order of magnitude lower than the 26 nM binding observed for *wt* SpaA$_{SLH}$ (Table 1 and Supplementary Fig. 7).

To confirm the structural basis for this reduced affinity, we determined the crystal structure of SpaA$_{SLH}$/G109A alone and in complex with 4,6-Pyr-β-D-ManNAcOMe. Unliganded SpaA$_{SLH}$/ G109A is similar in structure to *wt* SpaA$_{SLH}$ (Cα rmsd of 0.34 Å, excluding residues 44–54), and the Gly109Ala mutation is accommodated with no change in the position of Gln108 or Ala109 backbone atoms compared to Gln108 and Gly109 in the *wt* structure, but with a minor shift of residues 111–114. When SpaA$_{SLH}$/G109A crystals were soaked with 4,6-Pyr-β-D-ManNA-cOMe, no electron density for ligand was observed in the resulting structures. We then co-crystallized SpaA$_{SLH}$/G109A with ligand but attempts to solve the structure by molecular replacement using *wt* SpaA$_{SLH}$ structures were unsuccessful. Instead, we solved the structure using SAD phasing from a KI-soaked co-crystal.

The resulting structure of SpaA$_{SLH}$/G109A in complex with 4,6-Pyr-β-D-ManNAcOMe reveals a stunning structural change. While no electron density for ligand is observed in the G2 pocket, indicating that the Gly109Ala mutation inactivates binding as hypothesized, the protein is observed to alter significantly its conformation to utilize the G1 pocket instead, where excellent electron density for ligand is observed (Figs. 4 and 5, and Supplementary Fig. 6). The conformational change involves the movement of SLH2 away from SLH3 toward SLH1, resulting in a widening of G2 from 8.1 to 11.9 Å (measured between Trp151 Cη2 and Ala109 O) and a narrowing of G1 from 8.9 to 7.6 Å (between Trp93 Cη2 and Gly46 O). The overall Cα rmsd between liganded and unliganded SpaA$_{SLH}$/G109A is 1.73 Å (1.52 Å when excluding residues 44–54), and when the SLH3 domain helices are aligned there is an 8.5 Å maximum displacement of SLH2 Cα atoms (measured for Lys91; Fig. 5). This significant structural change explains the failure to observe ligand soaked into crystals of unliganded SpaA$_{SLH}$/G109A, because the structural change required for binding would not be possible in the crystal context, and also explains the initial failure to solve the complex structure by molecular replacement due to the significant difference between the search model and target structure. Ligand binding in G1 of SpaA$_{SLH}$/G109A occurs with the SLH-28 and SLH-Gly29 backbone flip observed through residues Ser45 and Gly46 (Figs. 4 and 5, and Supplementary Table 2), again suggesting that the conservation of SLH-Gly29 is for its role in this binding mechanism.

To demonstrate conclusively the significance of the conserved SLH-Gly29 and the backbone flip for SCWP binding, we created a double mutant SpaA$_{SLH}$/G46A/G109A for binding and structural analysis. As expected, this mutant displays no detectable binding to 4,6-Pyr-β-D-ManNAcOMe by ITC (Table 1 and Supplementary Fig. 7). The crystal structure of SpaA$_{SLH}$/G46A/G109A

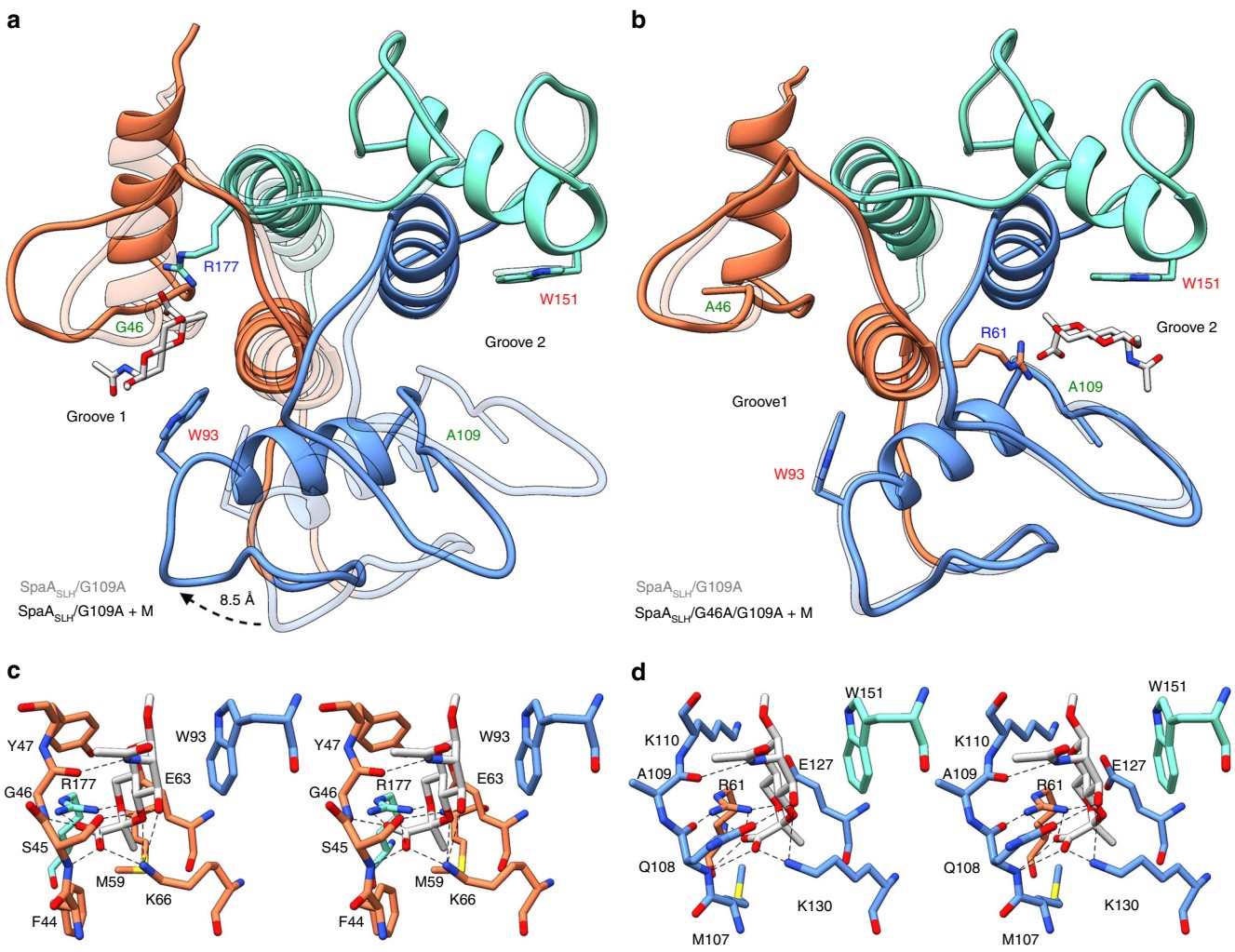

**Fig. 5** SpaA$_{SLH}$/G109A ligand binding and conformational change. Ribbon diagrams of **a** SpaA$_{SLH}$/G109A with bound monosaccharide overlayed with unliganded SpaA$_{SLH}$/G109A (transparent), and **b** SpaA$_{SLH}$/G46A/G109A with bound monosaccharide overlayed with unliganded SpaA$_{SLH}$/G109A (transparent). Alignments were performed using the helices of SLH3. Stereo views of monosaccharide binding in G1 of SpaA$_{SLH}$/G109A and G2 of SpaA$_{SLH}$/G46A/G109A are shown in **c** and **d**

displays the same groove conformations as *wt* SpaA$_{SLH}$ and unliganded SpaA$_{SLH}$/G109A, with G2 narrower than G1 (Fig. 5b). Residues 46–55 in G1 are in a similar conformation as in the unliganded SpaA$_{SLH}$/G109A structure, but with a slight widening of the loop to accommodate the Gly46Ala mutation. Despite the lack of binding detected by ITC, ligand is again observed bound in the G2 pocket, albeit with Gln108 and Ala109 in the unliganded backbone conformation (i.e., not flipped) (Fig. 5d). The monosaccharide is bound with the same interactions as observed for *wt* SpaA$_{SLH}$, except that there is no hydrogen bond between the pyruvyl carboxyl oxygen and Gly109 nitrogen, but instead there is a minor movement of ligand away from the Gln108 carbonyl that results in a less ideal bidentate interaction with Arg61. The observation of bound ligand in the crystal structure despite no binding being detected by ITC, similar to the observation of 2:1 binding in the *wt* SpaA$_{SLH}$ crystal structure despite 1:1 binding measured by ITC, indicates again that affinity is markedly increased in the crystal context.

**SLH-Gly29 is required for cellular S-layer anchoring**. To corroborate the structural and biophysical evidence for the importance of the SLH-Gly29 backbone flip to SCWP binding, we investigated the phenotypic effects of SLH-Gly29 mutation in live

cells. Unfortunately, cellular experiments with SpaA are impossible because genetic manipulation of the *spaA* gene in *P. alvei* results in a lethal phenotype. Instead, we probed a different cell surface protein, SlhA, which also possesses three SLH domains with variations of the conserved TRAE and GIIxG motifs (Fig. 1). It was shown previously that deletion of the *P. alvei slhA* gene produces changes of the colony morphology, impedes biofilm formation, and results in loss of swarming motility of *P. alvei* cells[49]. Swarming is a known mechanism for the migration of cells or cell clusters on semi-solid surfaces produced by the movement of flagella[50].

SlhA possesses the conserved SLH-Gly29 of the SLH domain profile (Fig. 1) in domains SLH2 and SLH3, but not SLH1, and possesses the conserved SLH-Arg43 of the TRAE motif and SLH-Trp13 (or Tyr) in all three domains. Presuming the same arrangement of these conserved residues in three binding grooves as observed for SpaA and Sap (Fig. 3), it is likely that SlhA possesses two functional SCWP binding sites in G2 and G3. Therefore, we created a double mutant of SlhA with both Gly1224 and Gly1293 mutated to alanine (corresponding to SLH-Gly29 in SLH2 and SLH3) for cellular testing.

To assess the phenotypic effect of SLH-Gly29 mutation in SlhA, we used a *P. alvei ΔslhA* knockout strain, which shows loss

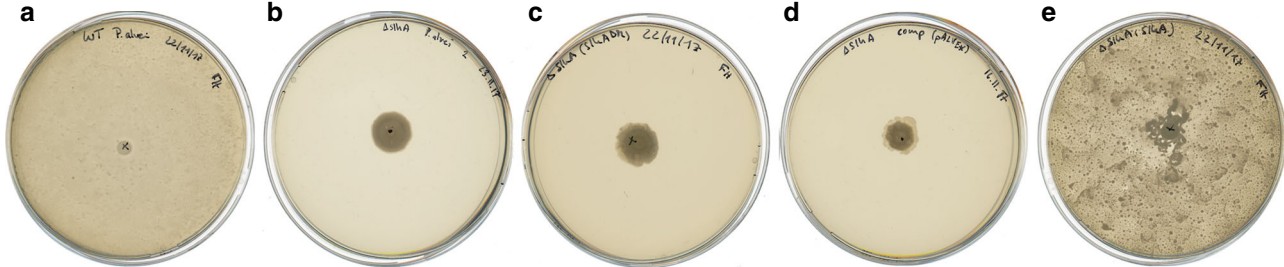

**Fig. 6** Swarming motility of *P. alvei* CCM 2051[T] variants. LB agar plates showing the growth of **a** *P. alvei wt*, **b** *P. alvei ΔslhA*, **c** *P. alvei ΔslhA* complemented with *slhA/G1224A/G1293A*, **d** *P. alvei ΔslhA* complemented with empty plasmid pEXALV, and **e** *P. alvei ΔslhA* complemented with *slhA*

of swarming motility on semi-solid agar plates compared to *wt P. alvei* (Fig. 6), and transformed it with plasmids encoding *wt* SlhA or the SlhA/G1224A/G1293A double mutant. Transformation with plasmid encoding SlhA/G1224A/G1293A resulted in a similar phenotype as transformation with an empty plasmid or as the knockout strain, indicating the loss of swarming motility. Transformation with plasmid encoding *wt* SlhA restored swarming motility in *P. alvei ΔslhA*, as demonstrated previously by Janesch et al.[49]. These results suggest that *P. alvei* loses the ability to swarm because SlhA/G1224A/G1293A is unable to attach to the cell surface as a consequence of the mutation of the conserved SLH-Gly29 residues to alanines, and thus confirm the biological importance of the SLH-Gly29 backbone flip to SCWP binding.

**Structural basis for inequality within SpaA SLH domains**. Previous mutagenesis studies on full-length *P. alvei* CCM 2051[T] SpaA have shown that mutations of the TRAE, TVEE, and TRAQ motifs to TAAA have unequal impacts, where they result in 37%, 88%, and 50% of *wt* binding to cell wall sacculli, respectively[27]. This inequality of tandem SLH domains in cell wall binding can now be correlated to the complex structures reported here.

The TRAE motif of SLH1 contains Arg61, which forms direct contacts with the pyruvate moiety of 4,6-Pyr-β-D-ManNAcOMe in the primary binding pocket in G2 (Fig. 3). Glu63 contributes to the secondary binding pocket in G1 and interacts with Arg177 but not with bound ligand (Figs. 3 and 5). The TRAE/TAAA mutation therefore affects both the G1 and G2 sites, consistent with the lowest binding observed when these residues were mutated (37% of *wt*)[27].

The TVEE motif of SLH2 lacks the conserved SLH-Arg43, and instead presents Val125 to the non-functional site in G3. Glu127 interacts with Arg61 in the primary binding pocket of G2, but, as observed for Glu63 of the TRAE motif, does not interact with bound ligand (Fig. 3). Given the minor decrease in SCWP binding upon mutation of TVEE to TAAA (88% of *wt*)[27], a structural role of Glu127 in the G2 pocket is not critical to binding.

The TRAQ motif of SLH3 contains Arg177, which forms direct contacts with the pyruvate moiety of 4,6-Pyr-β-D-ManNAcOMe in the secondary binding site in G1 (Figs. 2 and 3). Gln179 of the TRAQ motif lies in the non-functional site in G3. The contribution of the TRAQ motif to only the secondary binding site in G1 explains the lesser impact of the TRAQ/TAAA mutation (50% of *wt*) than the TRAE/TAAA mutation (37% of *wt*)[27].

**Comparison of SpaA_SLH to other SCWP-binding domains**. The only other published structures of SLH domains are those from the *B. anthracis* S-layer protein Sap, which were published in the unliganded form by Kern et al. (PDB 3PYW)[39], and in complex with a synthetic trisaccharide 4,6-Pyr-β-D-ManNAc-(1 → 4)-β-D-

GlcNAc-(1 → 6)-α-GlcN (approximating the terminal unit of *B. anthracis* SCWP) by Sychantha et al. during the revision of our manuscript (PDB 6BT4)[41]. Sychantha et al. report ligand binding only in G2 and hypothesize that binding in G1 and G3 are obstructed by crystal packing. The terminal 4,6-Pyr-β-D-ManNAc of ligand is bound in G2 with O6 and a pyruvyl carboxyl coordinated by Arg72 (corresponding to SLH-Arg43) and with stacking interactions to Trp164 (corresponding to SLH-Trp13), as observed for 4,6-Pyr-β-D-ManNAcOMe bound by SpaA_SLH, while the base of the binding pocket about the pyruvyl methyl is formed by different hydrophobic residues (Supplementary Fig. 9). Ligand is rotated by ~20° away from SLH-Arg43 in Sap_SLH compared to SpaA_SLH so that Trp164 stacks against the 1 → 4 glycosidic linkage.

Sychantha et al. report that the pyruvyl carboxyl groups of ligand form hydrogen bonds with both the backbone amide and carbonyl oxygen of Lys117 (corresponding to SLH-28). This backbone orientation suggests that the Gly118 (corresponding to SLH-Gly29) backbone amide does not hydrogen bond to ligand and that the backbone flip does not occur with binding in this groove. However, inspection of the associated electron density reveals that the Lys117 backbone carbonyl is modeled outside of 2Fo–Fc density, and there is Fo–Fc density that supports instead a flipped orientation of the peptide bond (Supplementary Fig. 9). When flipped, Gly118 adopts backbone torsions disallowed for all other amino acids, and has its amide positioned to hydrogen bond with the pyruvyl moiety of ligand. This supports our hypothesis that the role of the conserved SLH-Gly29 is to allow this backbone orientation for ligand binding. Unfortunately, the structure of unliganded Sap_SLH[39] displays ambiguous electron density for this region, so it cannot be concluded that SLH-Gly29 also allows a stabilizing interaction between the SLH-28 backbone carbonyl and SLH-Arg43 in the unliganded state in this case. However, G1 of the unliganded Sap_SLH structure does display well-defined electron density for this loop, and the backbone carbonyl of SLH-28 hydrogen bonds with SLH-Arg43, and SLH-Gly29 is positioned to perform a backbone flip upon ligand binding.

Although binding was observed only in G2 of the Sychantha et al. structure, both G1 and G3 contain conserved SLH-Trp13, SLH-Gly29, and SLH-Arg43 (Lys in G1) residues that were found to indicate active binding sites in SpaA_SLH. Significantly, both G1 and G3 of Sap_SLH are in "open" conformations, as observed for G1 of SpaA_SLH, compared to the "closed" conformations observed for the ligand-bound G2 of SpaA_SLH or G1 of SpaA_SLH/G109A (Supplementary Fig. 9). This indicates that if G1 or G3 of Sap_SLH is capable of binding SCWP, it would require significant structural changes to narrow these grooves, as observed for SpaA_SLH/G109A. Indeed, Sychantha et al. report that crystals would crack and dissolve when soaked with increasing ligand concentrations, which suggests that conformational changes are occurring in the protein to accommodate binding in alternate

(perhaps even preferred) sites. Given the complexity of SCWP binding by *P. alvei* SpaA$_{SLH}$ elucidated by our study, where binding in a secondary site in G1 was accomplished by a significant structural change after deactivating the primary binding site in G2 by SLH-Gly29Ala mutation, it is clear that further investigation is required into the potential activity and cooperativity of the multiple grooves in *B. anthracis* Sap$_{SLH}$.

Besides the SLH domain, the only other known S-layer anchoring module in Gram-positive bacteria is the cell wall binding 2 (CWB2) domain, prevalent in *Clostridia*[30]. The recent crystal structures of the *Clostridium difficile* 630 cell wall proteins Cwp6 and Cwp8 (PDBs 5J72 and 5J6Q) revealed trimeric arrangements of three tandem CWB2 domains that, like SLH domains, possess a parallel three-helix bundle core, but otherwise bear little resemblance[33]. On the sides of the inner three-helix bundle of the CWB2 trimer, there are additional sheets and helices that occupy the spaces where the SLH domain trimer has three grooves and SCWP-binding sites. The binding site for the CWB2 ligand PS-II, an anionic teichoic acid-like polymer consisting of hexa-glycosyl phosphate repeats, was hypothesized to be a shallow V-shaped groove on the surface of the trimer perpendicular to the core helix bundle that outlines the second CWB2 domain. Docking studies placed one hexa-glycosyl phosphate repeat lying in either direction in this groove, with the terminal mannose-1-P phosphate moiety placed near conserved arginine residues[33]. However, it remains possible that the CWB2 domain instead recognizes primarily the terminal mannose-1-P in a manner analogous to the recognition of the terminal pyruvylated residue of SCWP by SLH domains as seen for SpaA$_{SLH}$ and Sap$_{SLH}$. Together, the trimeric arrangement of CWB2 and SLH domains and the use of conserved arginine residues to recognize anionic moieties of diverse cell wall polysaccharides may support a common evolutionary origin of these cell wall anchoring modules in Gram-positive bacteria.

## Discussion

The crystal structures and binding analyses presented here of *P. alvei* SpaA$_{SLH}$, SpaA$_{SLH}$/G109A, and SpaA$_{SLH}$/G46A/G109A with synthetic SCWP fragments provide novel insights into the functional contributions of many conserved SLH domain residues, including SLH-Trp13, SLH-Gly29, and SLH-Arg43, which precisely interlock from three different SLH domains to generate two active binding sites. These SLH domains appear to be specific for the non-reducing-end Pyr-β-D-ManNAc moiety of *P. alvei* SCWP, as opposed to internal occurrences of Pyr-β-D-ManNAc with β-D-GlcNAc-(1 → 3) linkages. This specificity is consistent with that of *B. anthracis* Sap$_{SLH}$ for its SCWP that is pyruvylated only on the terminal ManNAc[40,41], which may indicate a general trend in the recognition of terminal anionic moieties of SCWPs by SLH domains in other organisms. However, this raises further questions as to why *P. alvei* SCWP is pyruvylated on every repeating unit, or why some SCWP of other organisms are otherwise negatively charged on every repeating unit, rather than on only the terminal moieties utilized for anchoring.

The most highly conserved SLH domain residue SLH-Gly29 is shown to allow a backbone flip of residues SLH-28 and SLH-29 that is critical for SCWP binding in both biophysical and cellular experiments. This backbone flip appears to have evolved as an elegant dual-purpose mechanism that promotes proper folding and stability of the SLH domain trimer in the unliganded state and readjusts to achieve optimal ligand coordination in the bound state. Given the high conservation of SLH-Gly29, we envision that this mechanism applies to the recognition of pyruvylated SCWP by SLH domains in other organisms, and perhaps even other non-pyruvylated anionic SCWPs. Indeed, this backbone flip is supported by the electron density of the recently published structure of *B. anthracis* Sap$_{SLH}$ in complex with 4,6-Pyr-β-D-ManNAc-(1 → 4)-β-D-GlcNAc-(1 → 6)-α-D-GlcN (PDB 6BT4)[41].

Our characterization of two functional SCWP-binding sites in SpaA$_{SLH}$ reveals a sequence pattern that may predict the functional-binding sites of other SLH domain trimers; a binding site in the groove beside domain SLH$_n$ is predicted by the presence of SLH-Gly29 in SLH$_n$ followed by SLH-Trp13 in SLH$_{n+1}$ and SLH-Arg43 in SLH$_{n+2}$ (Figs. 1 and 3), or circular permutations thereof. Indeed, it appears that many proteins containing SLH domain trimers identified in the Pfam family SLH (PF00395) contain either two or three copies of this sequence pattern. However, it should be noted that large structural changes mediating binding between sites may conceal the activity of some sites from analysis, as demonstrated here for *P. alvei* SpaA$_{SLH}$.

When ligand binding in SpaA$_{SLH}$ G2 was disrupted by the SLH-Gly29Ala mutation, we observed a significant domain movement that restructured G1 to allow binding. The 1:1 binding to 4,6-Pyr-β-D-ManNAcOMe with 29 nM $K_D$ measured for *wt* SpaA$_{SLH}$ corresponds to a primary site in G2, while the 1:1 binding with 226 nM $K_D$ measured for SpaA$_{SLH}$/G109A corresponds to a secondary site in G1. SpaA$_{SLH}$ therefore appears to display a novel form of intramolecular negative cooperativity, where two mutually exclusive binding sites for the same ligand are modulated by a significant structural change. Although binding in G1 could only be detected when G2 was deactivated by the SLH-Gly29Ala mutation, the relationship of the two sites in a biological context may be more fluid. In the paracrystalline protein S-layer, there would be significant crowding between SLH domains of neighboring S-layer proteins and the interspersed SCWP strands, which may stabilize binding in a manner similar to the crystallization-induced binding of monosaccharide in G1 of SpaA$_{SLH}$ that was not detected in solution by ITC. Indeed, this crowding in the S-layer context may induce domain movement and the exchange of active binding sites, especially at locations of S-layer strain. Switchable binding between two sites may thus have biological significance in allowing relocation of binding among different SCWP strands to alleviate strain caused by S-layer restructuring during cell growth or division or could facilitate movement of SLH-anchored proteins on the cell surface. It remains possible that the switchable binding observed for SpaA$_{SLH}$ is an exception rather than the rule, and that the SLH domain trimer evolved instead for avid binding and has lost functionality in some cases according to the avidity required for certain SCWPs, or for the function of the particular protein being anchored. However, the crystal structure of *B. anthracis* Sap$_{SLH}$ reveals one "closed" and two "open" grooves, each possessing the conserved residues that predict active SCWP-binding sites[39,41], which suggests that it may utilize such a switchable binding mechanism as well.

Given the broad biological importance of S-layers in the survival and virulence of diverse microorganisms, the mechanisms by which these proteins attach to cell surfaces are potential targets for the development of novel antibiotics. This is foreshadowed by the lethality of *spaA* inactivation in *P. alvei* and by our demonstration that swarming motility is impaired by disrupting SLH domain anchoring of the S-layer protein SlhA. Our in-depth characterization of the SpaA$_{SLH}$–SCWP interaction in the model organism *P. alvei* should serve as a benchmark for future studies wishing to elucidate or perhaps inhibit such interactions in other organisms.

## Methods

**Expression and purification of recombinant SpaA$_{SLH}$.** The SpaA$_{SLH}$ protein comprising amino acids 21–193, including a Ser-Gly-Ser linker preceding a C-terminal His$_6$-tag, was generated by standard PCR amplification from *P. alvei* CCM

2051[T] genomic DNA. Primers for PCR amplification are listed in Supplementary Table 3. The PCR product and pET22b vector were digested with NdeI and SacI and ligated. The resulting plasmid, pET22b-spaAslh, was transformed into E. coli BL21 (DE3-Star) cells (Invitrogen) and plated on LB agar plates containing 100 µg mL$^{-1}$ of ampicillin. Single colonies were transferred to 5 mL of LB medium supplied with the antibiotic and grown overnight at 37 °C. This culture was inoculated into four flasks containing 0.5 L of ampicillin-containing LB medium and incubated at 37 °C under shaking (200 rpm). Protein overexpression was induced at an OD$_{600}$ ~0.6 by the addition of IPTG at a final concentration of 0.6 mM. Cells were further incubated for 3 h, harvested by centrifugation at 6000 × g for 20 min and pellets were stored at −20 °C.

Cell pellets were resuspended in lysis buffer (25 mM Tris-HCl, pH 8.0, 200 mM NaCl, 5 mM imidazole) and disrupted by sonication. The lysate was clarified by centrifugation and applied to a nickel NTA-affinity chromatography column (Qiagen) equilibrated with lysis buffer. The column was washed with ten column volumes of buffer A (25 mM Tris-HCl, pH 8.0, 20 mM imidazole), followed by ten volumes of buffer B (25 mM Tris-HCl, pH 8.0, 50 mM imidazole). His$_6$-tagged SpaA$_{SLH}$ was eluted with 25 mM Tris-HCl, pH 8.0, 250 mM imidazole (elution buffer). Further purification of SpaA$_{SLH}$ was performed by size exclusion chromatography on a Superdex 75 16/60 column equilibrated with 20 mM HEPES, pH 7.5, containing 100 mM NaCl. Fractions were analyzed by SDS-PAGE (10% gel) and proteins were detected by Coomassie brilliant blue (CBB).

**Site-directed mutagenesis.** Sited-directed mutagenesis of residues glycine 109 and glycine 46 to alanine were performed by overlapping extension PCR to generate SpaA$_{SLH}$/G109A (single mutant) and SpaA$_{SLH}$/G46A/G109A (double mutant). The upstream and downstream part of the mutation site in SpaA$_{SLH}$ were amplified separately using as templates the purified recombinant plasmids pET22b-spaA$_{SLH}$ for the single mutant and pET22b-spaA$_{SLH}$/G109A for the double mutant. The forward primer of the downstream part and the reverse primer of the upstream part were overlapping and included the point mutations that were introduced in both elongations. These two amplicons were mixed and amplified in a second round of PCR. Template DNA was degraded by DpnI. Nicked plasmids were transformed into E. coli DH5α. The presence of G46A and G109A mutations was confirmed by direct DNA sequencing. Expression and purification of the sinlge and double mutant proteins were performed as described for the SpaA$_{SLH}$ variant (see above). Primers for site-directed mutagenesis are listed in Supplementary Table 3.

**Generation of double TAAA mutants.** SpaA$_{SLH}$/TAAA$_{12}$ and SpaA$_{SLH}$/TAAA$_{13}$ mutants were generated by standard PCR amplification with the primers Fwd-SpaA$_{SLH}$-NdeI and Rev-SpaA$_{SLH}$-SacI utilized to generate SpaA$_{SLH}$ (residues 21–93), using as template plasmid pETSMut4H (pET28a carrying His$_6$-tagged spaA mutated in TRAE and TVEE motifs) for SpaA$_{SLH}$/TAAA$_{12}$ and plasmid pET-Mut5H (pET28a carrying His$_6$-tagged spaA mutated in TRAE and TRAQ motifs) for SpaA$_{SLH}$/TAAA$_{13}$[27]. The PCR products were digested and ligated into linearized pET22b vector using the NdeI and SacI restriction enzymes. The presence of the TAAA mutations in SpaA$_{SLH}$/TAAA variants was confirmed by direct DNA sequencing. Expression and purification of SpaA$_{SLH}$/TAAA$_{12}$ and SpaA$_{SLH}$/TAAA$_{13}$ mutants were carried out as described for the SpaA$_{SLH}$ variant.

**Generation of mutant P. alvei CCM 2051[T] strains.** In a first round, the SlhA$_{SLH}$ protein comprising amino acids 1124–1335 (the three consecutive SLH domains of SlhA) was generated by standard PCR amplification from P. alvei CCM 2051[T] genomic DNA. The PCR product was ligated into the pET22b vector using the NdeI and XhoI restriction sites. Subsequently, site-directed mutagenesis of residues glycine 1224 and glycine 1293 of protein SlhA to alanine was performed by overlapping extension PCR. The upstream and downstream part of the mutation sites were amplified separately using as templates the purified, recombinant plasmids pET22b-slhA$_{SLH}$ for the single mutant and pET22b-slhA$_{SLH}$/G1293A for the double mutant (for more details, see section Site-directed mutagenesis). The presence of G1224 and G1293 mutations was confirmed by direct DNA sequencing. Next, a DNA fragment comprising around 400 bps, where the two glycine point mutations were included, was amplified by standard PCR using as template the purified, recombinant plasmid pET22b-slhA$_{SLH}$/G1224A/G1293A. The amplification product was ligated into linearized vector pEXALV_P(SlhA)_SlhA[49], which bears slhA native promotor, utilizing the PvuII and KpnI restriction sites. Primers for PCR amplification and site-directed mutagenesis are listed in Supplementary Table 3. The resulting plasmid, pEXALV_P(SlhA)_SlhA/G1224A/G1293A, was transformed into P. alvei CMM 2051[T] ΔslhA cells as described by Zarschler et al.[51]. Briefly, 500 ng of plasmid DNA was added to an aliquot of electro-competent cells, then the cell suspension was transferred into a pre-cooled 1-mm electroporation cuvette. After application of the pulse, the cell suspension was diluted with 4 mL of pre-warmed casein–peptone soymeal–peptone broth, containing 250 mM sucrose, 5 mM MgCl$_2$, and 5 mM MgSO$_4$. Cells were incubated for 2 h at 37 °C, and then platted on LB agar plates supplemented with 10 µg/mL chloramphenicol and incubated overnight at 37 °C. The electroporation conditions were 200 Ω/25 µF/20 kV cm$^{-1}$.

**Synthesis of SCWP ligands.** Described in Supplementary Methods.

**Circular dichroism spectroscopy.** CD spectroscopy for folding and secondary structure analyses of SpaA$_{SLH}$ was performed on a Chirascan CD apparatus (Applied Photophysics, Leatherhead, UK) equipped with a thermostatic cell holder and a Peltier element for temperature control. The instrument was flushed with a nitrogen flow at a rate of 5 L min$^{-1}$ and measurements were performed in the far UV region (180–260 nm) (Supplementary Fig. 7). The instrument parameters were as follows: path length, 1.0 mm; spectral bandwidth, 3.0 nm; step size, 1.0 nm; scan period, 10 s. Spectra were baseline-corrected to remove birefringence of the quartz cell. SpaA$_{SLH}$ was analyzed at a concentration of 10 µM in 20 mM KH$_2$PO$_4$, pH 7.0, at 20 °C.

To analyze the thermal stability of SpaA$_{SLH}$, single wavelength scans were performed at 208 nm using the same protein solution as above, with a scan time per point of 10 s and stepwise temperature increase rate of 1 °C min$^{-1}$ over an interval from 20 to 90 °C (Supplementary Fig. 7).

**Isothermal titration calorimetry.** ITC was performed using a MCS titration calorimeter (Microcal, Inc., Northampton, MA). Concentrations of the monosaccharide and disaccharide ligands were determined based on dry weight and confirmed by high-performance anion exchange chromatography with pulsed electrochemical detection on a PA-1 column (Thermo Fisher)[52]. Protein concentration was determined by UV/Vis light spectroscopy using a molar extinction coefficient of 18450 M$^{-1}$ cm$^{-1}$ at 280 nm. To avoid sample-related artifacts, protein solutions were freshly prepared prior to each set of titration experiments by dialysis of SpaA$_{SLH}$, SpaA$_{SLH}$/G109A, and SpaA$_{SLH}$/G46A/G109A, respectively, against 20 mM KH$_2$PO$_4$, pH 7.0. The buffer dialysate was used for concentration adjustments and blank titrations. ITC measurements were done at 20 °C in 20 mM KH$_2$PO$_4$, pH 7.0. Protein, ligand, and buffer solutions were degassed prior to running the titrations. The sample cell was filled with 0.020 mM SpaA$_{SLH}$ solution (0.25 mL), the reference cell was filled with buffer and the injection syringe contained 20-times concentrated ligand solution. The injection sequence consisted of an initial injection of 0.5 µL of ligand solution to prevent low binding enthalpies arising from the filling of the syringe, followed by injection of 1 µL of ligand solution at 150-s intervals, each, until complete saturation of SpaA$_{SLH}$-binding sites had been reached. SpaA$_{SLH}$/G109A and SpaA$_{SLH}$/G46A/G109A 0.020 mM solutions were titrated with a 20-times concentrated monosaccharide ligand solution, following a similar titration sequence as with wt SpaA$_{SLH}$. A blank sample was run in the absence of protein to determine the heat of dilution of the ligand. Data analysis was done with Origin software (Microcal Inc.) by fitting a single-site binding isotherm. The obtained thermodynamic parameters were enthalpy of binding ΔH°, entropy of binding ΔS°, and the association constant $K_a$. Measurements were performed at least in triplicate. Binding isotherms are shown in Supplementary Fig. 7.

**Crystallization and ligand soaking.** Recombinant, purified SpaA$_{SLH}$ was concentrated to 30 mg mL$^{-1}$ using Amicon ultra centrifugal filter units. Crystal screens were prepared using an Art Robbins Instruments crystal gryphon robot and Hampton 96-well Intelli plates. Crystals were obtained with the Hampton Index screen in condition number 54 (50 mM calcium chloride dihydrate, 0.1 M bis–Tris, pH 6.5, and 30% v/v PEG MME 550), and optimized by hanging drop vapor diffusion in 35 × 10 mm tissue culture dishes in the same condition at 16 °C with higher protein-to-reservoir ratios. Crystals grew within 1–3 days. 4,6-Pyr-β-D-ManNAcOMe was soaked into existing crystals of SpaA$_{SLH}$ at a concentration of 10 mM. 4,6-Pyr-β-D-ManNAcOMe was also co-crystallized with SpaA$_{SLH}$ in the same crystallization condition at 5 mM and 27 mg mL$^{-1}$, respectively. Additional co-crystals with 4,6-Pyr-β-D-ManNAcOMe were obtained from the Hampton Index screen, with the crystals that provided the C2 structure grown in condition 66 (0.2 M ammonium sulfate, 0.1 M bis–Tris, pH 5.5, and 25% w/v PEG 3350), and the P1 structure in condition 87 (0.2 M sodium malonate, pH 7.0, 20% w/v PEG 3350). Co-crystals of SpaA$_{SLH}$ and β-D-GlcNAc-(1 → 3)-4,6-Pyr-β-D-ManNAcOMe were obtained in Hampton Index screen condition number 57 (50 mM ammonium sulfate, 50 mM bis–Tris, pH 6.5, and 30% v/v pentaerythritolethoxylate [15/4 EO/OH]). Crystals of SpaA$_{SLH}$/G109A were obtained with the Hampton Index screen in condition number 81 (0.2 M ammonium acetate, 0.1 M Tris, pH 8.5, and 25% w/v PEG 3350). Co-crystals of SpaA$_{SLH}$/G109A with 4,6-Pyr-β-D-ManNAcOMe were obtained with the Hampton Index screen in condition number 95 (0.1 M potassium thiocyanate, pH 6.8, and 30% PEG MME 2000). Co-crystals of SpaA$_{SLH}$/G46A/G109A with 4,6-Pyr-β-D-ManNAcOMe were obtained with the Hampton Index screen in condition number 47 (0.1 M bis–Tris, pH 6.5, and 28% PEG MME 2000).

**Data collection, structure determination, and refinement.** X-ray diffraction data were collected either on a Rigaku R-AXIS IV++ area detector with X-rays produced by a Rigaku MM-003 generator, on a Marmosaic CCD300 detector at beamline CMCF-ID at the Canadian Light Source (CLS) synchrotron (Saskatoon, SK), or on a Dectris Pilatus 200 K detector with X-rays produced by a Rigaku Micromax-007 HF generator. Data were scaled, averaged, and integrated using HKL2000[53]. Crystals of unliganded SpaA$_{SLH}$ and co-crystals of SpaA$_{SLH}$ with 4,6-Pyr-β-D-ManNAcOMe were soaked overnight at 16 °C in mother liquor with the addition of 0.2 M KI prior to data collection. Phenix AutoSol[54] was used to solve

SAD phases from iodide ions and to generate initial models. Additional data sets were collected from native crystals and solved by molecular replacement with Phaser[55] using the lower-resolution KI derivative structures as search models. Data sets from SpaA$_{SLH}$ crystals soaked and co-crystallized with synthetic 4,6-Pyr-β-D-ManNAcOMe, from co-crystals of SpaA$_{SLH}$ with β-D-GlcNAc-(1 → 3)-4,6-Pyr-β-D-ManNAcOMe, from unliganded SpaA$_{SLH}$/G109A, and from co-crystals of SpaA$_{SLH}$/G46A/G109A with 4,6-Pyr-β-D-ManNAcOMe were solved by molecular replacement using the structure of unliganded SpaA$_{SLH}$ as a search model. All model building and refinement were carried out using Coot[56] and Refmac5 through the CCP4 interface[57,58].

**Visualization and graphics**. Figure 1 was produced using PROSITE sequence logo for domain profile PS51272[59]. The chemical structures in Fig. 2 were prepared using MarvinSketch (version 15.11.30.0; http://chemaxon.com). All protein structure and electron density figures were produced with UCSF Chimera. Chimera is developed by the Resource for Biocomputing, Visualization, and Informatics at the University of California, San Francisco (supported by NIGMS P41-GM103311)[60].

**Swarming motility assay**. To investigate the role of the SLH-Gly29 residue in SLH domain-carrying proteins of Gram-positive bacteria within a biological context, an assay based on *P. alvei* CCM 2051$^T$ swarming ability on agar plates was used[49]. To test the swarming motility of *P. alvei* Δ*slhA* and *P. alvei* Δ*slhA*$_{comp}$ strains provided with plasmid-encoded *slhA* and *slhA*/G1224A/G1293A in comparison to *P. alvei wt*, cells were grown overnight (OD$_{600}$ ~2.0) and 5 µL of each culture were applied on 1% (semi-solid) LB agar plates. *P. alvei wt* and *P. alvei* Δ*slhA* cells were incubated at 37 °C for 24 h, whereas complemented, pEXALV vector-based *P. alvei* strains were incubated at 37 °C for 48 h. Images were taken using an EPSON PERFECTION V750 PRO scanner. The experiments were repeated at least in triplicates.

**Data availability**. The atomic coordinates and structure factors (Supplementary Table 1) have been deposited in the Protein Data Bank under the accession codes 6CWC, 6CWF, 6CWH, 6CWI, 6CWL, 6CWM, 6CWN, and 6CWR. The data that support the findings of this study are available from the corresponding author on request.

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

## Acknowledgements

The authors thank Dr. Paul Furtmüller and Dr. Roland Ludwig for assistance with CD and ITC measurements, Dr. Andreas Hofinger-Horvath for recording the NMR spectra, and Dr. Jean-Baptiste Farcet for providing MS data. This work was supported by the Austrian Science Fund FWF, projects P24305-B20 (to P.M.) and P27374-B22 (to C.S.), and by the Natural Sciences and Engineering Research Council of Canada (to R.J.B., S.M.L.G. and S.V.E.).

## Author contributions

R.J.B., A.L.-G., F.F.H., G.M., P.K., S.M.L.G., O.H.-G. and B.J. designed and performed the experiments. P.K., P.M., C.S. and S.V.E. supervised the project. R.J.B., A.L.-G., F.F.H., B. J., G.M., P.K., P.M., C.S. and S.V.E. contributed to writing the paper.

## Additional information

**Competing interests:** The authors declare no competing interests.

