## [Peer Review File · Nature Communications]

Reviewers' comments:

Reviewer #1 (Remarks to the Author):

This manuscript by Blackler et al describes the complete structure of the *P. alvei* SLH domain in complex with its secondary cell wall polysaccharide ligand. The SLH domain is the most widespread mechanism of S-layer anchoring in the Gram positive bacteria. This structure provides the first insight into the molecular basis of SLH function.

Overall I have no criticisms of the experimental design or execution. However there are some important omissions in background context and interpretation that I feel must be addressed for the sake of clarity in the scientific record.

The first description of the interaction between SLH domains and a pyruvylated SCWP was Mesnage et al (2000) EMBO J 19:4473-4484. Despite that study focusing on the S-layer of *B. anthracis*, it seems very strange that this paper has not been cited.

The structure of the *B. anthracis* SLH domain has also been previously described by the Schneewind lab. The only reference to this structure in the introduction is a passing comment on page 4. Although the structure is mentioned again at the beginning of the results, no detailed description is given. A more rounded comparison of this structure with that published previously should be added for completeness. I would suggest that a supplemental figure comparing the structures should be added at a minimum.

Several of the extended data figures do not appear to be mentioned in the text.

The conclusions section in the main text is rather superficial. If this is due to a space constraint, more of the methods could surely be moved to supplemental. An expanded conclusions section could better explain the wider implications of the current study.

For example:

How much of the current data is relevant to the wider family of SLH-containing proteins?
Does analysis of other SLH domains support the two mutually exclusive binding sites model?
More broadly, how does this structure compare to the recently published *C. difficile* CWB2 domain structure (Usenek et al (2017) Structure 25:1-8)? Although these structures are

apparently distinct modules for S-layer anchoring, the structures appear to be strikingly similar.

Reviewer #2 (Remarks to the Author):

The manuscript by Blackler and co-workers describes a careful crystallographic study performed with the SLH domains of the S-layer protein SpaA, in unbound form as well as associated to mimics of secondary cell wall polymers. Proteins involved in S-layer formation are interesting targets for the development of new, potential antibiotics, and work in this field is thus timely and of interest.

SpaA from the strain *Paenibacillus alvei* was thus selected for this work, and the objective was to employ X-ray crystallography and isothermal titration calorimetry to perform an extended investigation on the binding characteristics between SpaA and secondary cell wall polymers. Despite the fact that the structure of the SHL region of SpaA reported here is new, SLH domain proteins have been crystallized and their structures have been reported. Of note is the structure of Sap from *B. anthracis*, solved by Joachimiak and Schneewind and referred to in the manuscript, whose fold is highly reminiscent of the structure presented here.

Blackler and colleagues investigate the structure of SpaA in complex with a synthetic monosaccharide by solving the crystal structure of the complex in different space groups, and discuss the interactions observed in detail. In most cases, the ligand is bound in pocket G2, with the exception of the structure solved in space group C2, where they could observe some density in G1. Using ITC, they identify that the interaction between the monosaccharide and SpaA (in G2) occurs with nanomolar affinity, and relate binding in the G1 pocket to a potential artifact. They continue their work by solving the structure in the presence of a disaccharide, which binds in G2 similarly to the monosaccharide. The most interesting observation involves a local glycine flip that occurs upon ligand binding within one of the important motifs. In order to characterize the flip, they created glycine-to-alanine single and double mutants, and characterized them by crystallography and ITC. The authors then discuss the structural contributions of different conserved residues to ligand binding.

The manuscript describes crystal structures that have been solved and refined carefully, and the local modifications engendered by ligand binding are certainly of interest for specialists in the field. However, in the absence of any microbiological or cellular data indicating how these results could be relevant for control of bacterial survival or novel inhibitor development, I believe that the paper would be more appropriate for a specialized structural biology journal. Authors should also ensure that labels are clearly visible, especially in figures that involve several superimpositions, such as Figs. 2, 3, and 4.

Reviewer #3 (Remarks to the Author):

The manuscript presented here by Blackler et. al. attempts to understand the role of SLH domains in the cell-wall of Gram-positive bacteria. Overall the manuscript is interesting, scientifically sound, and provides an extremely detailed look at how these domains interact with the cell-wall. An understanding of the S-layer and the cell-wall in general is not only academically fascinating but a critical step towards the creation of new antibiotics.

Overall based on scientific merit and importance this is a very nice study. The manuscript could benefit from some re-writing to better explain and put into context various experiments and hypothesis, for example a more in depth introduction and a more extended discussion. Currently, it is unclear what certain terms are or why certain experiments were performed unless you are familiar with the field (see comments).

Additionally, the authors have made some very interesting structural observations that may result directly to the biology of how SLH proteins interact with the S-layer. In particular, the study of the conserved glycine-motifs is very interesting but requires in vivo experimentation to prove.

The recommendation is to revise the manuscript before the final decision.

Major Comments

The authors have shown biochemically and structurally that the glycine residues in the conserved GIIxG motif are an important part of the ligand binding mechanism. However, they do not cite any literature where these mutations have been tested in bacteria. Is this experiment possible, or are there papers they can cite? If this is a completely new finding, the authors should test the glycine point mutations in their species (if possible) but at a minimum the equivalent mutations in another Gram-positive bacteria, such as *B. subtilis*, that is routinely studied genetically.

Page 11, Mutual-exclusive... I do not believe the conclusion here is justified. First, the authors use ITC to state that G1 binding is artificial, then they make mutants to show that binding in G1 is possible. They then conclude this is a novel form of intramolecular negative cooperativity. This section is best saved for the conclusions and discussed in a more hypothetical manner. For example, they could only observe a significant structural change by mutating a highly conserved residue. How do the authors show that it's not just an artifact of manipulating the system? Their conclusion/hypothesis here about the dual-binding in the context of the S-layer is intriguing and very possibly correct, but it hasn't been proven.

This paper definitely requires more discussion. A single paragraph seems insufficient.

Minor Comments

Summary

“Here we represent the high-resolution...” change “the” to “a”

“interaction, with complex crystal structures...” remove “complex” change to co-crystal, “S-layer protein SpaA and defined” change to “protein SpaA in complex with defined..”

“The SpaASLH pseudo-trimer” Maybe re-write to emphasize the biological importance first then the technological or drug application? Also maybe state that SpaA is observed to oligomerize – “pseudo trimer” needs to be explained to the reader before its used.

Introduction

Page 3. “there are over 54,000 specific hits...” What type of hits? The SLH domain? What was the search based upon exactly?

Page 4. “SLH domains and SCWP not possible previously.” Switch possible and previously.

Page 4, last paragraph. What is mean by Inequality within other SLH domain repeats? Also, it is a better read if the last paragraph of the introduction highlights or at least previews one or two major points from the paper.

The introduction seems very short. Maybe explain the system better? For example, when it gets to the results its not explained why they used certain sugars for the co-crystal structures and biochemical characterization. It is eluded to, but when it comes up maybe state exactly why each ligand was used (eg. the Pyr is critical as mentioned, GlnNAc, ManNAc – why? – its non-obvious to the general reader).

Results

Page 5, first paragraph. Maybe compare to the B. anthracis Sap? Is there anything biologically important from their similarities and differences? For example, the SLH domains and the G1-3 grooves, are they exactly the same as Sap?

Page 5, first paragraph. Maybe make it clear that Val125 of TVEE is in an equivalent position as R, but due to the residue change doesn't have the same contacts? Is that what the authors meant to imply?

Page 5, last paragraph. "unliganded trigonal structures..." only a crystallographer is going to understand what you mean – maybe reference the table and PDB files here as well?

Page 6, first paragraph. Is this loop specifically not involved in any crystal packing contacts in any of the structures? What are the b-factors like in the 44-55 loop? Are they high relative to the mean of the structure?

Page 6, second paragraph. The authors need to place a starting sentence that explains why they performed ITC to address what problem. It seems just dropped in here without context. Additionally, the authors cannot say with 100% certainty that this is an artifact. They should put a qualifier "most likely, we believe, etc."

Did the authors perform the control of buffer into protein? If so, it should be added to the extended data figure.

Page 7, last paragraph. The glycine flip is really interesting, however a table is not sufficient. The authors should show structural examples with the electron density of the region to show the flip.

Page 8, "To test this hypothesis, the SpaA/G109A" change "the" to "a". Additionally, the ligand still binds but is close to an order of magnitude worse in affinity, correct? 26 nM vs. 226 nM? The authors should state/say something about this.

Page 8, middle paragraph. "The unliganded crystal structure" The phrasing needs reworking.

Page 9, first paragraph. Please reference the extended data figure for the ITC.

"Interestingly, despite..." This is a long run-on sentence. Consider revising.

Page 9, Inequality within tandem. Please be sure that inequality is defined and its clear what this means, as per previous comment.

Page 12, conclusions. "These crystal structures..." Consider re-phrasing? "The crystal structures presented here..."

I would like to express my gratitude to the Reviewers for their careful review. Below are the Reviewers' comments reproduced verbatim, with our responses interspersed.

Reviewer #1 (Remarks to the Author):

This manuscript by Blackler et al describes the complete structure of the *P. alvei* SLH domain in complex with its secondary cell wall polysaccharide ligand. The SLH domain is the most widespread mechanism of S-layer anchoring in the Gram positive bacteria. This structure provides the first insight into the molecular basis of SLH function.

Overall I have no criticisms of the experimental design or execution. However there are some important omissions in background context and interpretation that I feel must be addressed for the sake of clarity in the scientific record.

COMMENT 1: The first description of the interaction between SLH domains and a pyruvylated SCWP was Mesnage et al (2000) EMBO J 19:4473-4484. Despite that study focusing on the S-layer of *B. anthracis*, it seems very strange that this paper has not been cited.

The structure of the *B. anthracis* SLH domain has also been previously described by the Schneewind lab. The only reference to this structure in the introduction is a passing comment on page 4. Although the structure is mentioned again at the beginning of the results, no detailed description is given. A more rounded comparison of this structure with that published previously should be added for completeness. I would suggest that a supplemental figure comparing the structures should be added at a minimum.

RESPONSE 1.1: The omission of the Mesnage et al (2000) EMBO citation was an oversight. This paper is now referenced in the introduction.

We have also added a more detailed description of the published structure of *B. anthracis* Sap_{SLH} to the introduction, and a new results section titled "Comparison of Spa_{ASLH} to other SCWP-binding domains" that includes a thorough comparison (including a supplementary figure).

COMMENT 1.2: Several of the extended data figures do not appear to be mentioned in the text.

RESPONSE 1.2: All supplementary figures and tables are now referenced in the text.

COMMENT 1.3: The conclusions section in the main text is rather superficial. If this is due to a space constraint, more of the methods could surely be moved to supplemental. An expanded conclusions section could better explain the wider implications of the current study.

For example:

How much of the current data is relevant to the wider family of SLH-containing proteins? Does analysis of other SLH domains support the two mutually exclusive binding sites model? More broadly, how does this structure compare to the recently published *C. difficile* CWB2 domain structure (Usenek et al (2017) Structure 25:1-8)? Although these structures are apparently distinct modules for S-layer anchoring, the structures appear to be strikingly similar.

RESPONSE 1.3: The reviewer is correct that space constraints were the reason for the brief conclusions section, as the manuscript was initially prepared for *Nature* with a stricter word limit. In reformatting for *Nature Communications*, we have now appropriately revised and expanded the Conclusions (now Discussion). An additional results section “Comparison of Spa_{SLH} to other SCWP-binding domains” includes thorough comparisons to the recent structures of *B. anthracis* Sap_{SLH} and *C. difficile* CWB2 domain-containing proteins Cwp6 and Cwp8.

Interestingly, the structure of Sap_{SLH} in complex with a synthetic SCWP ligand published during our revisions (Sychantha *et al.* 2018, *Biochemistry*. DOI: 10.1021/acs.biochem.8b00060) revealed ligand bound only in G2, with G2 in a ‘closed’ conformation and G1 and G3 in ‘open’ conformations. In our study, wild-type Sap_{SLH} was also observed with ligand bound in a ‘closed’ G2 with G1 and G3 ‘open’, while Sap_{SLH}/G109A was observed instead with ligand bound in a ‘closed’ G1 with G2 and G3 ‘open’.

Both G1 and G3 of Sap_{SLH} contain conserved residues involved in SCWP binding, which suggests Sap_{SLH} may also utilize a switchable binding mechanism. However, it is important to note that Sychantha *et al.* do not hypothesize this binding mechanism, and that they have completely overlooked the function of the conserved SLH-Gly29 and have modelled it incorrectly in their deposited data. Therefore, the findings and discussion presented in our study remain completely novel.

Reviewer #2 (Remarks to the Author):

The manuscript by Blackler and co-workers describes a careful crystallographic study performed with the SLH domains of the S-layer protein SpaA, in unbound form as well as associated to mimics of secondary cell wall polymers. Proteins involved in S-layer formation are interesting targets for the development of new, potential antibiotics, and work in this field is thus timely and of interest.

SpaA from the strain *Paenibacillus alvei* was thus selected for this work, and the objective was to employ X-ray crystallography and isothermal titration calorimetry to perform an extended investigation on the binding characteristics between SpaA and secondary cell wall polymers. Despite the fact that the structure of the SHL region of SpaA reported here is new, SLH domain proteins have been crystallized and their structures have been reported. Of note is the structure of Sap from *B. anthracis*, solved by Joachimiak and Schneewind and referred to in the manuscript, whose fold is highly reminiscent of the structure presented here.

Blackler and colleagues investigate the structure of SpaA in complex with a synthetic monosaccharide by solving the crystal structure of the complex in different space groups, and discuss the interactions observed in detail. In most cases, the ligand is bound in pocket G2, with the exception of the structure solved in space group C2, where they could observe some density in G1. Using ITC, they identify that the interaction between the monosaccharide and SpaA (in G2) occurs with nanomolar affinity, and relate binding in the G1 pocket to a potential artifact. They continue their work by solving the structure in the presence of a disaccharide, which binds in G2 similarly to the monosaccharide. The most interesting observation involves a local glycine flip that occurs upon ligand binding within one of the important motifs. In order to characterize the flip, they created glycine-to-alanine single and

double mutants, and characterized them by crystallography and ITC. The authors then discuss the structural contributions of different conserved residues to ligand binding.

The manuscript describes crystal structures that have been solved and refined carefully, and the local modifications engendered by ligand binding are certainly of interest for specialists in the field.

COMMENT 2.1: However, in the absence of any microbiological or cellular data indicating how these results could be relevant for control of bacterial survival or novel inhibitor development, I believe that the paper would be more appropriate for a specialized structural biology journal.

RESPONSE 2.1: We have performed additional *in vivo* studies that prove the conserved SLH-Gly29 is required for SCWP binding and cell-surface protein anchoring in a biological context. Unfortunately, direct *in vivo* experiments with SpaA are impossible because genetic manipulation of the *spaA* gene in *P. alvei* results in a lethal phenotype. Therefore, to address the reviewers' concerns, we probed a different SLH domain-containing cell surface protein, SlhA. It was shown previously that deletion of the *P. alvei* *slhA* gene produces changes in colony morphology and impedes biofilm formation and swarming motility of *P. alvei* cells. We now show that two single-point SLH-Gly29Ala mutations in the *slhA* gene produces the same phenotype as gene deletion, thus indicating a loss of cell-surface anchoring and proving the significance of this conserved residue in a biological context.

S-layers are critical for the survival and virulence of diverse microorganisms, and we believe that our detailed elucidation of a conserved anchoring mechanism, now supported by *in vivo* data using a second *P. alvei* SLH domain-carrying surface protein, will be of broad interest and significance.

COMMENT 2.2: Authors should also ensure that labels are clearly visible, especially in figures that involve several superimpositions, such as Figs. 2, 3, and 4.

RESPONSE 2.2: All figures have been revised and/or replaced for clarity.

Reviewer #3 (Remarks to the Author):

The manuscript presented here by Blackler et. al. attempts to understand the role of SLH domains in the cell-wall of Gram-positive bacteria. Overall the manuscript is interesting, scientifically sound, and provides an extremely detailed look at how these domains interact with the cell-wall. An understanding of the S-layer and the cell-wall in general is not only academically fascinating but a critical step towards the creation of new antibiotics.

Overall based on scientific merit and importance this is a very nice study. The manuscript could benefit from some re-writing to better explain and put into context various experiments and hypothesis, for example a more in depth introduction and a more extended discussion. Currently, it is unclear what certain terms are or why certain experiments were performed unless you are familiar with the field (see comments).

Additionally, the authors have made some very interesting structural observations that may result directly to the biology of how SLH proteins interact with the S-layer. In particular, the study of the conserved glycine-motifs is very interesting but requires *in vivo* experimentation to prove.

The recommendation is to revise the manuscript before the final decision.

Major Comments

COMMENT 3.1: The authors have shown biochemically and structurally that the glycine residues in the conserved GIIxG motif are an important part of the ligand binding mechanism. However, they do not cite any literature where these mutations have been tested in bacteria. Is this experiment possible, or are there papers they can cite? If this is a completely new finding, the authors should test the glycine point mutations in their species (if possible) but at a minimum the equivalent mutations in another Gram-positive bacteria, such as *B. subtilis*, that is routinely studied genetically.

RESPONSE 3.1: The involvement of SLH-Gly29 and the GIIxG motif in SCWP binding is a novel finding of our study, and mutations of this motif have not been assessed previously to the best of our knowledge. As discussed in Response 2.1, we performed additional *in vivo* studies that prove the conserved SLH-Gly29 is required for SCWP binding in a biological context, and these results are included in the revised manuscript. With regard to testing SLH-Gly29 mutations in other species; assessing SLH-Gly29 mutations in *B. subtilis* is not possible as there are no SLH domain-containing proteins identified in that organism. Switching to an additional new system, where we do not know the structure of the cell wall ligand, let alone have available pure, synthesized partial structures of the ligand would, we feel, be well beyond the scope of this study.

COMMENT 3.2: Page 11, Mutual-exclusive... I do not believe the conclusion here is justified. First, the authors use ITC to state that G1 binding is artificial, then they make mutants to show that binding in G1 is possible. They then conclude this is a novel form of intramolecular negative cooperativity. This section is best saved for the conclusions and discussed in a more hypothetical manner. For example, they could only observe a significant structural change by mutating a highly conserved residue. How do the authors show that it's not just an artifact of manipulating the system? Their conclusion/hypothesis here about the dual-binding in the context of the S-layer is intriguing and very possibly correct, but it hasn't been proven.

RESPONSE 3.2: Based on 1:1 binding with 29 nM K_D measured by ITC for wild-type SpaA_{SLH} binding to the SCWP monosaccharide 4,6-Pyr- β -D-ManNAcOMe, and the consistent observation of ligand bound in G2 of crystal structures, we did conclude that the single observation of ligand bound with fragmented electron density in G1 of wild type SpaA_{SLH} was an artifact of crystallization. However, when binding in G2 was disrupted by SLH-Gly29Ala mutation, we observed a significant structural change that allowed binding in G1 with 226 nM K_D . Unfortunately, it is impossible for us to prove or disprove that G1 is functional in wild-type SpaA_{SLH} because G2 has higher affinity for ligand and binding in G2 precludes the structural change required for binding in G1. However, we believe that the maintenance of highly conserved residues in G1 and its displayed 226 nM K_D suggest that it does have biological significance. Nevertheless, we acknowledge that this section is largely hypothetical and have moved it to the discussion as recommended by the reviewer.

COMMENT 3.3: This paper definitely requires more discussion. A single paragraph seems insufficient.

RESPONSE 3.3: As mentioned for Reviewer 1, the length was a result of the manuscript being initially submitted to *Nature*. The manuscript has been significantly revised and now includes an expanded discussion.

Minor Comments

Summary

COMMENT 3.4: “Here we represent the high-resolution...” change “the” to “a”

RESPONSE 3.4: Done

COMMENT 3.5: “interaction, with complex crystal structures...” remove “complex” change to co-crystal, “S-layer protein SpaA and defined” change to “protein SpaA in complex with defined..”

RESPONSE 3.5: Done

COMMENT 3.6: “The SpaASLH pseudo-trimer” Maybe re-write to emphasize the biological importance first then the technological or drug application? Also maybe state that SpaA is observed to oligomerize – “pseudo trimer” needs to be explained to the reader before its used.

RESPONSE 3.6: The term ‘pseudo-trimer’ has been replaced by “three SLH domains” or “SLH domain trimer” for clarity.

Introduction

COMMENT 3.7: Page 3. “there are over 54,000 specific hits...” What type of hits? The SLH domain? What was the search based upon exactly?

RESPONSE 3.7: This statement originally referred to sequences identified using NCBI’s conserved domain database (CDD) as “specific hits”, defined as those sequences found by RPS-BLAST to contain the domain “with an E-value that is equal to or lower than a domain-specific Threshold E-value”. We have revised this statement to give more curated statistics from the EMBL Pfam database, i.e. the number of sequence matches and bacterial species in UniProt reference proteomes.

COMMENT 3.8: Page 4. “SLH domains and SCWP not possible previously.” Switch possible and previously.

RESPONSE 3.8: Sentence was revised.

COMMENT 3.9: Page 4, last paragraph. What is mean by Inequality within other SLH domain repeats? Also, it is a better read if the last paragraph of the introduction highlights or at least previews one or two major points from the paper.

RESPONSE 3.9: The penultimate paragraph was revised to include an experimental example describing inequality. The final paragraph was revised to preview major findings.

COMMENT 3.10: The introduction seems very short. Maybe explain the system better? For example, when it gets to the results its not explained why they used certain sugars for the co-crystal structures and biochemical characterization. It is eluded to, but when it comes up maybe state exactly why each ligand was used (eg. the Pyr is critical as mentioned, GlnNAc, ManNAc – why? – its non-obvious to the general reader).

RESPONSE 3.10: Introduction was expanded to explain the system better, including descriptions of SCWPs and SLH domain structures. A new figure (Fig. 2) was added that shows schematics of selected SCWPs and the corresponding ligands used in the study.

Results

COMMENT 3.11: Page 5, first paragraph. Maybe compare to the *B. antracis* Sap? Is there anything biologically important from their similarities and differences? For example, the SLH domains and the G1-3 grooves, are they exactly the same as Sap?

RESPONSE 3.11: An overlay of SpaA_{SLH} and Sap_{SLH} structures was added to Fig 3 for initial general comparison, and a detailed comparison was added in the new results section “Comparison of SpaA_{SLH} to other SCWP-binding domains” and in the new supplemental Figure 5.

COMMENT 3.12: Page 5, first paragraph. Maybe make it clear that Val125 of TVEE is in an equivalent position as R, but due to the residue change doesn't have the same contacts? Is that what the authors meant to imply?

RESPONSE 3.12: That is correct. We revised this sentence for clarity: “In the case of SLH2, Val125 of the TVEE motif corresponds in position to the conserved SLH-Arg43 of the TRAE and TRAQ motifs but does not protrude into the neighboring G3.”

COMMENT 3.13: Page 5, last paragraph “unliganded trigonal structures...” only a crystallographer is going to understand what you mean – maybe reference the table and PDB files here as well?

RESPONSE 3.13: All structures are now referred to by PDB ID.

COMMENT 3.14: Page 6, first paragraph. Is this loop specifically not involved in any crystal packing contacts in any of the structures? What are the b-factors like in the 44-55 loop? Are they high relative to the mean of the structure?

RESPONSE 3.14: we have added a sentence about crystal contacts and B-factors of this loop, and added Supplementary Fig. 1 showing multiple conformations colored by B-factor.

COMMENT 3.15: Page 6, second paragraph. The authors need to place a starting sentence that explains why they performed ITC to address what problem. It seems just dropped in here without context. Additionally, the authors cannot say with 100% certainty that this is an artifact. They should put a qualifier “most likely, we believe, etc.”

RESPONSE 3.15: Paragraph was revised to clarify the incentive for ITC and qualify the hypothesis.

COMMENT 3.16: Did the authors perform the control of buffer into protein? If so, it should be added to the extended data figure.

RESPONSE 3.16: We did not perform the control of buffer into protein, but we did perform the control of non-pyruvylated ManNAc into protein, which showed no binding. This isotherm was added to Supplementary Fig. 3.

COMMENT 3.17: Page 7, last paragraph. The glycine flip is really interesting, however a table is not sufficient. The authors should show structural examples with the electron density of the region to show the flip.

RESPONSE 3.17: Structures with electron density showing the glycine flip are presented in the new Fig 5.

COMMENT 3.18: Page 8, “To test this hypothesis, the SpaA/G109A” change “the” to “a”. Additionally, the ligand still binds but is close to an order of magnitude worse in affinity, correct? 26 nM vs. 226 nM? The authors should state/say something about this.

RESPONSE 3.18: The reviewer is correct. This sentence has been revised for clarity (now page 10): “ITC revealed 1:1 binding to monosaccharide 4,6-Pyr- β -D-ManNAcOMe with 226 nM affinity, nearly an order of magnitude lower than the 26 nM binding observed for *wt* SpaA_{SLH}.”

COMMENT 3.19: Page 8, middle paragraph. “The unliganded crystal structure” The phrasing needs reworking.

RESPONSE 3.19: Revised: “To confirm the structural basis for this reduced affinity, we determined the crystal structure of SpaA_{SLH}/G109A alone and in complex with 4,6-Pyr- β -D-ManNAcOMe. Unliganded SpaA_{SLH}/G109A is similar in structure...”

COMMENT 3.20: Page 9, first paragraph. Please reference the extended data figure for the ITC.

RESPONSE 3.20: Done

COMMENT 3.21: “Interestingly, despite...” This is a long run-on sentence. Consider revising.

RESPONSE 3.21: Revised: “Despite the lack of binding detected by ITC, ligand is again observed bound in the G2 pocket, albeit with Gln108 and Ala109 in the unliganded backbone conformation (i.e. not flipped).”

COMMENT 3.22: Page 9, Inequality within tandem. Please be sure that inequality is defined and its clear what this means, as per previous comment.

RESPONSE 3.22: Revised paragraph for clarity.

COMMENT 3.23: Page 12, conclusions. “These crystal structures...” Consider re-phrasing? “The crystal structures presented here...”

RESPONSE 3.23: Revised: “The crystal structures and binding analyses presented here...”

Reviewers' Comments:

Reviewer #1 (Remarks to the Author):

This is a welcome resubmission of the manuscript by Blackler et al describing the molecular basis of the interaction between SLH domains and pyruvylated SCWPs.

The manuscript has been clearly improved and I thank the authors for addressing all of my comments. I have no further concerns and I believe that publication of this paper will be a valuable addition to the S-layer and Gram positive cell wall protein literature.

Reviewer #2 (Remarks to the Author):

The authors have adequately addressed my main point, which concerned the absence of microbiological data that extended their observations, and have performed swarming motility studies on wild type and mutant strains. My only further comment regarding this matter is that the experiments presented are not considered 'in vivo', which is a term generally employed for work involving animal models, etc, but rather 'in cellulo' or simply microbiologic. In order to be considered as bona fide 'in vivo' experiments, it is this reviewer's opinion that the manuscript would have had to include animal infection data.

Reviewer #3 (Remarks to the Author):

The authors have addressed all my concerns and I think it's a great paper and nicely presented.

In my opinion it suitable for publication in Nature Communications.

My only final suggestion would be in Figure 5 to label some residues to help orient the reader.

RESPONSES TO REVIEWERS' COMMENTS:

Reviewer #2 (Remarks to the Author):

The authors have adequately addressed my main point, which concerned the absence of microbiological data that extended their observations, and have performed swarming motility studies on wild type and mutant strains. My only further comment regarding this matter is that the experiments presented are not considered 'in vivo', which is a term generally employed for work involving animal models, etc, but rather 'in cellulo' or simply microbiologic. In order to be considered as bona fide 'in vivo' experiments, it is this reviewer's opinion that the manuscript would have had to include animal infection data.

Response: The term 'in vivo' has been removed from the manuscript and replaced with appropriate alternatives.

Reviewer #3 (Remarks to the Author):

The authors have addressed all my concerns and I think it's a great paper and nicely presented.

In my opinion it suitable for publication in Nature Communications.

My only final suggestion would be in Figure 5 to label some residues to help orient the reader.

Response: Residue labels have been added to Fig. 5.